# ATP-Binding Cassette Transporter Family C Protein 10 Participates in the Synthesis and Efflux of Hexosylceramides in Liver Cells

**DOI:** 10.3390/nu14204401

**Published:** 2022-10-20

**Authors:** Jahangir Iqbal, Meghan T. Walsh, M. Mahmood Hussain

**Affiliations:** 1Department of Cell Biology, SUNY Downstate Medical Center, Brooklyn, NY 11203, USA; 2King Abdullah International Medical Research Center, King Saud bin Abdulaziz University for Health Sciences, Ministry of National Guard Health Affairs, Al Ahsa 31982, Saudi Arabia; 3Department of Foundations of Medicine, NYU Long Island School of Medicine, Mineola, NY 11501, USA

**Keywords:** ABCA1, sphingolipids, HDL, glucosylceramide, ceramide, sphingomyelin, efflux

## Abstract

In addition to sphingomyelin and ceramide, sugar derivatives of ceramides, hexosylceramides (HexCer) are the major circulating sphingolipids. We have shown that silencing of ABCA1 transmembrane protein function for instance in cases of loss of function of ABCA1 gene results in low levels of HDL as well as a concomitant reduction in plasma HexCer levels. However, proteins involved in hepatic synthesis and egress of HexCer from cells is not well known although ABCA1 seems to be indirectly controlling the HexCer plasma levels by supporting HDL synthesis. In this study, we hypothesized that protein(s) other than ABCA1 are involved in the transport of HexCer to HDL. Using an unbiased knockdown approach, we found that ATP-binding cassette transporter protein C10 (ABCC10) participates in the synthesis of HexCer and thereby affects egress to HDL in human hepatoma Huh-7 cells. Furthermore, livers from ABCC10 deficient mice had significantly lower levels of HexCer compared to wild type livers. These studies suggest that ABCC10 partakes in modulating the synthesis and subsequent efflux of HexCer to HDL in liver cells.

## 1. Introduction

Sphingolipids play a critical role in physiology and are important components of cell membranes. They also act as bioactive lipids in several metabolic and hormonal pathways. The activities of enzymes in sphingolipid synthesis are regulated by fluxes of different substrates and through signaling in response to various stimuli [1,2,3]. Ample information is available about the synthesis, metabolism, and signaling activities of sphingolipids [4,5,6,7,8], but little is known about their transport to circulation.

Ceramide (Cer) is the simplest sphingolipid that is a precursor for the synthesis of complex sphingolipids such as sphingomyelin (SM) and hexosylceramides (HexCer) that include glucosylceramide (GlcCer) and galactosylceramide. HexCer serve as precursors for the synthesis of several complex glycosphingolipids [9] and are present in all cell types [10]. Inhibition of glycosphingolipids has been shown to ameliorate arterial stiffness and atherosclerosis in rabbits and mice [11].

Synthesis of sphingolipids begins at the cytosolic leaflet of the endoplasmic reticulum to generate the precursor Cer that is utilized for the synthesis of complex sphingolipids (Figure 1). Majority of the Cer is transported via vesicular transport or by the protein ceramide transfer protein from its site of synthesis in the endoplasmic reticulum to the Golgi apparatus where it is further modified to SM and glycosphingolipids [12]. During the synthesis of SM in the Golgi, Cer must flip towards lumenal side to serve as the substrate for SM synthases as the active site of these enzymes faces the lumen of the Golgi [13,14]. Similarly, synthesis of GlcCer from Cer and UDP-glucose occurs on the cytosolic side of the Golgi by glucosylceramide synthase [15], a transmembrane protein having its active site facing the cytosol [12,16]. GlcCer synthesis is an important first step for the synthesis of other glycosphingolipids and is regulated by different factors [17]. All the other glycosyltransferases involved in the synthesis of complex glycosphingolipids using GlcCer as a precursor are on the lumenal side of the Golgi and, therefore, necessitates the flipping of GlcCer from the cytosol to the lumen for further glycosylation. It is presumed that this translocation of GlcCer from the cytosol to lumen is mediated by multiple Golgi and *trans* Golgi network flippases [12,16,18,19]. Recently, Budani et al. [20] showed that multiple ABC transporters potentially act as GlcCer flippases and differentially control glycosphingolipids biosynthesis. It is likely that there are more proteins that regulate HexCer synthesis.

Plasma HexCer are possible biomarkers of disease progression. Li et al. [21] showed that plasma HexCer are high in chronic hepatitis C virus patients with severe fibrosis. They suggested HexCer (d18:1/12:0) as a potential diagnostic marker for severe hepatic steatosis. HexCer (16:1) is also a possible biomarker for disease progression in relapsing multiple sclerosis patients [22]. Furthermore, HexCer levels increase in drug-induced hepatic phospholipidosis [23]. Thus, studies about the origins of plasma HexCer may be useful in understanding progression of different diseases.

To date, a total of 49 different ATP binding cassette (ABC) transporters have been identified in humans [24,25,26]. ABCC1 and ABCA12 were shown to transport GlcCer in vitro [27] and in keratinocytes [10]. We recently demonstrated that ABCA1 is a regulator of plasma glycosphingolipids [28]. However, how HexCer are transported to plasma HDL is unknown. In this study, we used an unbiased siRNA screening against different transporter proteins and identified ABCC10 as an important player in the synthesis and export of HexCer in Huh-7 cells.

ABCC10, also called multidrug resistance protein 7 (MRP7), imparts drug resistance in tumor cells to taxanes [29], such as paclitaxel and docetaxel, which are commonly used in the treatment of common cancers. In addition, ABCC10 might also confer resistance to microtubule alkaloids and certain nucleoside analogues. ABCC10 is predicted to have several substrates with a diverse range including peptides, drugs, phospholipids and sphingolipids. Interestingly, ABCC10 has been shown to selectively recognize glucuronides such as 17-β-estradiol-(17-β-δ-glucuronide) and leukotriene C4 [30,31]. Hopper-Borge et al. [30] generated ABCC10 deficient mice and showed that these mice succumb to early death when exposed to taxanes compared to wild type mice. These studies revealed that ABCC10 affords protection against paclitaxel-induced bone marrow toxicity. Using these mice, we show that ABCC10 deficiency reduces hepatic HexCer levels.

## 2. Materials and Methods

### 2.1. Materials

C6-NBD ceramide (6-((*N*-(7-Nitrobenz-2-oxa-1,3-diazol-4-yl)amino)caproxyl)sphin- gsine, catalog #6224) was from Setareh Biotech. *N*-palmitoyl-D-erythro-sphingosine [palmitoyl-1-[^14^C]-ceramide (C16-[^14^C]-ceramide, 50–60 mCi/mmol, 1.85–2.22 GBq/mmol, catalog #ARC 0831-50) was purchased from American Radiolabeled Chemicals, Inc. All other chemicals and solvents were from Fisher Scientific (Pittsburgh, PA, USA) or VWR International (Bridgeport, CT, USA).

### 2.2. Mice

The ABCC10 knockout mice on C57BL6/J background (31) were a kind gift from Dr. Elizabeth Hopper-Borge of Fox Chase Cancer Center, Philadelphia, PA. Mice were housed under a 12:12 h light: dark schedule and bred at Animal Bioresources (SUNY Downstate Medical Center) in accordance with all requirements of SUNY/NIH IACUC. Mice had ad libitum access to chow diet and water.

### 2.3. Quantification of Lipids

Plasma (100 µL) and liver (2 mg) from *Abcc10^+/+^* and *Abcc10^−/−^* mice (*n* = 4) mice were used to quantify HexCer and SM using HPLC-MS/MS [28,32].

### 2.4. Identification of Proteins Involved in HexCer Transport and Synthesis

Human hepatoma Huh-7 cells grown in Dulbecco’s modified Eagle’s medium (DMEM) supplemented with 10% fetal bovine serum (FBS), L-glutamine, and antibiotics, were reverse transfected with 50 nM *siCTRL* or siRNA in triplicate against various membrane transporters (Dharmacon, GE Life Sciences, Lafayette, CO, USA) using RNAiMAX (Thermo Fisher, catalog #13778150) for 72 h.

### 2.5. Dose Response Effect of siABCC10 on HexCer Export and Synthesis

Human hepatoma Huh-7 cells were reverse transfected with either 50 nM of *siCTRL* or different concentrations of *siABCC10* (12.5, 25, 50, and 100 nM) using RNAiMAX [33].

### 2.6. HexCer Egress and Synthesis

Huh-7 cells were incubated with 2 µM C6-NBD ceramide in DMEM supplemented with 10% FBS at 37 °C for 3 h. Next, cells were washed 3 times with DMEM plus 0.1% BSA. To study efflux, we added fresh DMEM containing 40 µg/mL of BSA or human serum HDL (catalog #MBS173147, MyBioSource, San Diego, CA, USA). In other experiments, Huh-7 cells were incubated with 0.2 µM [^14^C]-ceramide to study sphingolipid synthesis and export. Culture media were harvested and centrifuged (600 g, 2500 rpm, 4 °C, 15 min, Heraeus Fresco 21 Centrifuge, 75003424 rotor, ThermoFisher Scientific, Waltham, MA, USA) to pellet cells. Lipids were extracted from media and cells [34], dried and re-suspended (100 µL of isopropanol) for the separation of sphingolipids on thin layer chromatograph (TLC) silica plates (catalog #44931, Analtech, Inc., Newark, NJ, USA) using a CHCl_3_:CH_3_OH:C_6_H_5_CH_3_:NH_4_OH:H_2_O (40:40:20:0.4:1.6, ratios by volume) solvent system [28,32]. The TLC plates were scanned using a PhosphorImager (GE Healthcare, Chicago, IL, USA). Bands corresponding to Cer, SM and HexCer were quantified using ImageJ.

### 2.7. Cell-Free GCS Synthesis Assay

A buffer containing 50 mM Tris-HCl, 1 mM EDTA, 5% sucrose, pH 7.4 buffer containing a mixture of protease inhibitors was used to prepare liver homogenates from *Abcc10^+/+^* and *Abcc10^−/−^* mice. The homogenate (500 µg of protein) was added to a buffer containing 50 mM Tris-HCl, pH 7.4, 25 mM KCl, phosphatidylcholine (100 µg/mL), C6-NBD ceramide (3.3 µg/mL), and UDP-glucose (500 µM) in a total reaction volume of 200 µL and incubated for 1 h at 37 °C [35,36]. We added 200 µL of CHCl_3_/CH_3_OH (2:1, *v*/*v*) to stop the reaction. Lipids were extracted and used for TLC.

### 2.8. Statistics

Biochemical data between *Abcc10^+/+^* and *Abcc10^−/−^* mouse models and comparisons amongst different treatments in Huh-7 cells were evaluated using Student *t*-test (GraphPad Prism).

## 3. Results

### 3.1. Identification of ABCC10 as a HexCer Transporter

We showed previously that ABCA1 deficiency in humans and mice reduces plasma levels of HexCer [32]. However, induction, knockdown, inhibition or downregulation of ABCA1 in Huh-7 cells had no effect on the efflux of HexCer to HDL [32]. Hence, we concluded that ABCA1 may indirectly affect plasma HexCer levels by participating in the biogenesis of HDL. Next, we hypothesized that other unknown proteins exist that play important roles in the export of HexCer. To identify the HexCer transporter(s), we used the approach that we previously undertook to identify ABCA7 as an important regulator of SM biosynthesis and efflux [33]. Studies to identify the HexCer transporters were performed in parallel with those reported earlier in the identification of SM transporter [33]. Briefly, we knocked down ABC transporter family members using specific siRNAs in Huh-7 cells. These cells were then treated with C6-NBD ceramides and incubated with either bovine serum albumin (BSA) to quantify basal efflux (*siCTRL-BSA*) or with human HDL to study their transport to HDL (*siCTRL-HDL*) (Figure 2). HDL significantly increased HexCer efflux when compared to BSA (Figure 2A,B) consistent with our previous studies [32,33] that HDL enhances HexCer efflux. We observed that *siABCC10* significantly decreased HexCer export to HDL (Figure 2A,B). A second screening validated these results. *siCTRL* treated cells showed significantly increased HexCer efflux to HDL when compared to *siCTRL-BSA*, and this export was significantly reduced after *siABCC10* treatment (Figure 2C), suggesting that ABCC10 might participate in the egress of HexCer to HDL.

### 3.2. Dose–Response Effect of siABCC10 on HexCer Efflux and Synthesis

Treatment of Huh-7 cells with increasing concentrations of *siABCC10* followed by incubation with HDL for 3 h did not affect the export of Cer and SM, respectively (Figure 3A,B). Contrary to Cer or SM, efflux of HexCer to HDL was significantly reduced in a dose-dependent manner in *siABCC10-HDL* cells compared to *siCTRL-HDL* treated cells (Figure 3C). At 100 nM of *siABCC10*, HexCer efflux was reduced by ~67% (Figure 3C). Thus, treatment of cells with *siABCC10* is associated with reduced HexCer transport to HDL.

Next, we quantified the amounts of HexCer remained in the cells (Figure 3D–F). Increasing concentrations of *siABCC10* had no significant effect on the amounts of Cer and SM present in cells (Figure 3D,E). However, accumulation of HexCer in the cells was significantly reduced with increasing concentrations of *siABCC10* compared to siCTRL (Figure 3F) suggesting that ABCC10 may play a role in HexCer synthesis.

### 3.3. siABCC10 Decreases HexCer Synthesis in Hepatoma Cells

To evaluate the role of ABCC10 in HexCer synthesis, Huh-7 cells were transfected with *siCTRL* or *siABCC10* and incubated with C6-NBD ceramides for shorter time periods (0–60 min) and measured the synthesis of sphingolipids. *siABCC10* reduced ABCC10 mRNA (0.18 ± 0.04, *n* = 3) levels by 82% compared to *siCTRL* (1.00 ± 0.17, *n* = 3). Time dependent increases in cellular Cer levels were similar in both *siABCC10* and *siCTRL* treated cells (Figure 4A) indicating similar uptake. At the earliest time point in both *siCTRL* and *siABCC10* treated cells, NBD-Cer was converted to SM and its levels increased with time until 30 min. Subsequently, SM levels reached a similar steady state in *siCTRL* and *siABCC10* treated cells. These studies suggest that Cer uptake and its subsequent conversion to SM were not affected by ABCC10 deficiency. In contrast to SM, synthesis of HexCer showed time dependent increase with time in *siCTRL* and *siABCC10* treated cells and did not reach steady state during this time course. However, these temporal increases were significantly lower in *siABCC10* treated cells, thereby, leading to decreased intracellular HexCer levels compared to *siCTRL* cells (Figure 4A). These studies demonstrate that ABCC10 knockdown reduces HexCer synthesis but has no effect on SM synthesis.

We considered that reduced HexCer synthesis might be secondary to the use of NBD-labeled C6-Cer. As an additional approach, we labeled *siCTRL* and *siABCC10* treated Huh-7 cells with C16-[^14^C]-Cer and measured labeled media and cellular sphingolipids. We also knocked down ABCA1 whose deficiency reduces plasma HexCer [32]. We studied efflux of [^14^C]-sphingolipids to HDL and BSA (Figure 4B). There was a significant increase in Cer, SM and HexCer levels in the media obtained from *siCTRL-HDL* compared to *siCTRL-BSA* cells suggesting increased efflux to HDL. Compared to *siCTRL*, *siABCA1* had no effect on Cer and SM efflux to HDL but significantly increased HexCer efflux. Cer egress was significantly increased, whereas HexCer transport to HDL was significant decreased in *siABCC10-HDL* compared to *siCTRL-HDL* cells (Figure 4B). However, there was no significant difference in the egress of SM to HDL in *siABCC10-HDL* cells compared to *siCTRL-HDL* cells. These studies indicate that *siABCC10* significantly reduces [^14^C]-HexCer efflux to HDL without affecting SM efflux.

We then measured sphingolipids in cells (Figure 4C). *siCTRL-HDL* cells had reduced levels of Cer, SM and HexCer compared with *siCTRL-BSA* (Figure 4C) suggesting that increased transport (Figure 4B) is associated with lower cellular levels. *siABCA1-HDL* did not affect cellular HexCer levels compared to *siCTRL-HDL*. *siABCC10-HDL* had lower cellular Cer and SM levels compared to *siCTRL-HDL*, most likely secondary to their increased efflux to HDL. Since *siABCC10-HDL* showed decreased HexCer transport to HDL (Figure 4B), we had anticipated an increase in cellular levels of HexCer compared to *siCTRL-BSA* treated cells. However, our data showed lower levels of radiolabeled HexCer in *siABCC10-HDL* compared to *siCTRL-HDL* treated cells (Figure 4C). In short, *siABCC10-HDL* cells showed increased efflux of Cer and SM and reduced cellular levels. In contrast, *siABCC10-HDL* cells showed reduced efflux as well as diminished cellular levels. We interpret these data to suggest that ABCC10 deficiency reduces HexCer synthesis and, thereby, results in diminished export.

### 3.4. ABCC10 Deletion Decreases Hepatic HexCer Levels without Affecting Glucosylceramide Synthase Activity in Liver Homogenates

To determine whether ABCC10 plays a role in regulating physiological HexCer levels, we measured HexCer in the plasma and liver of wild type (*Abcc10^+/+^*) and ABCC10 knockout (*Abcc10^−/−^*) mice fed a chow diet (Figure 5A–D, Table 1). Deletion of ABCC10 in mice had no effect on total plasma SM or HexCer levels (Figure 5A,B, Table 1). Similarly, hepatic SM levels were unaffected by ABCC10 deficiency (Figure 5C, Table 1). However, we observed a significant decrease (~39%) in the levels of hepatic HexCer in *Abcc10^−/−^* as compared to *Abcc10^+/+^* mice (Figure 5D). Analysis of individual species of hepatic HexCer (Table 1) suggested a significant decrease in almost all the species of HexCer in *Abcc10^−/−^* mice, except for C22_1-HexCer, C22-HexCer and C26_HexCer. We also observed a significant decrease in the plasma levels of C18_1-HexCer, C18-HexCer, and C20-HexCer in *Abcc10^−/−^* mice (Table 1). Since these species were minor components of plasma HexCer, their individual reductions had no impact on total plasma HexCer (Figure 5B). These data indicate that ABCC10 regulates the levels of HexCer in the liver.

We considered the possibility that decreased levels of hepatic HexCer in ABCC10 knockout mice may be due to a reduction in the glucosylceramide synthase (GCS) activity. To evaluate this possibility, we conducted an in vitro cell-free assay to measure GCS activity, the major contributor of HexCer in liver cells, in the liver homogenates isolated from *Abcc10^+/+^* and *Abcc10^−/−^* mice [35,36]. In vitro incubation of NBD-Cer with UDP-glucose for 1 h without the addition of liver homogenates (No Hom) did not result in the synthesis of SM and GlcCer (Figure 5E). However, when the liver homogenates were added to the reaction mixture, there was an increase in the synthesis of both SM and GlcCer. Compared to the liver homogenates isolated from *Abcc10^+/+^* mice, we did not see any significant difference in the synthesis of either SM or GlcCer in the liver homogenates isolated from *Abcc10^−/−^* mice (Figure 5E). These data indicate that decrease in HexCer levels in ABCC10 knockout mice liver is likely not due to reductions in GCS activity.

## 4. Discussion

ABCA1 participates in the transport of cholesterol and phospholipids to HDL and its deficiency in humans and mice has been shown to associate with low levels of plasma HexCer [28]. However, mechanistic studies performed in Huh-7 cells revealed that ABCA1 is not directly involved in the efflux of HexCer. ABCA1 is critical in generating HDL that acts as a potent acceptor for the efflux of HexCer. In this study, using unbiased siRNA screening, we identified ABCC10 as a protein that plays a role in HexCer efflux (Figure 2). Our subsequent experiments showed that ABCC10 also participates in the biosynthesis of HexCer (Figure 3 and Figure 4). Thus, ABCC10 partakes in HexCer synthesis and, thereby, affects its efflux.

ABCC10 is expressed in several tissues [29,37,38,39] and has been shown to be present in plasma membrane as well as in intracellular compartments [40,41]. Intracellular localization of ABCC10 was unperturbed by various compounds used in these studies [40,41]. Expression of ABCC10 is elevated in hepatocellular carcinoma compared with adjacent healthy liver tissue [42]. ABCC10 expression is significantly higher in monocyte-derived macrophages and peripheral blood lymphocytes obtained from rheumatoid arthritis patients [43]. A bioinformatics analysis of genes expressed in samples of atherosclerotic lesions and control arteries without atherosclerotic lesions showed an increased expression of ABCC10 in lesion areas [44] suggesting a role of ABCC10 in atherosclerosis.

ABCC10 belongs to the ABCC family that partakes in drug elimination and efflux of other endogenous molecules [29,30,31,45]. Our cell culture studies indicate that ABCC10 plays a role in the biosynthesis and, thereby, in the efflux of HexCer. However, studies using liver homogenates demonstrated that ABCC10 deficiency has no effect on HexCer synthesis by GCS indicating that ABCC10 may not directly modulate the enzyme activity. We speculate that ABCC10 may modulate HexCer synthesis by affecting substrate availability and/or substrate removal from the GCS enzyme.

The overexpression of ABCC10 in HEK293 cells confers resistance to various chemotherapeutic drugs [29,30]. ABCC10 has also been shown to protect cells against paclitaxel toxicity [30]. Our current studies for the first time show that ABCC10 affects HexCer synthesis and export in Huh-7 cells. Thus, it is likely that ABCC10 is a multi-functional protein that plays a role in diverse physiologic functions.

The members of the ABC family are recognized players in the efflux of molecules and drugs. ABCA1 participates in the efflux of phospholipid and cholesterol [46,47]. We have recently shown that ABCA7 affects synthesis and export of SM and its deficiency in mice affects brain SM metabolism causing cognitive defects [33]. The current report indicates that ABCC10 plays a role in the synthesis of HexCer. Consistent with our studies, Budani et al. [20] identified several ABC transporters that may play roles as flippases and affect synthesis. These observations point out to the possibility that, besides being drug/molecule exporters, ABC transporters might be physiologically important modulators of sphingolipid biosynthesis.

HexCer consist of both GlcCer and GalCer. Our TLC approach of separating different sphingolipids does not resolve these two HexCer; therefore, it is unknown whether ABCC10 plays a role in the synthesis of both or not. Another drawback of this study is that we did not look at downstream changes in the biosynthesis of complex sphingolipids in ABCC10 deficient cells and mice.

In summary, we identified a novel role of ABCC10 in the biosynthesis and efflux of HexCer in human hepatoma cells. Furthermore, ABCC10 participates in controlling levels of several HexCer species in mouse liver. Thus, ABCC10 is a regulator of hepatic HexCer homeostasis.

## Figures and Tables

**Figure 1 nutrients-14-04401-f001:**
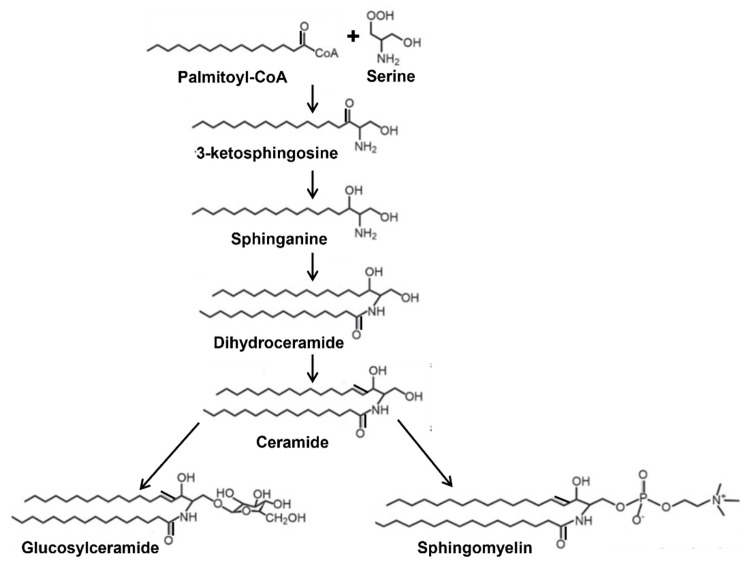
Simplified schematic representation of the synthesis of glucosylceramide and sphingomyelin from ceramide.

**Figure 2 nutrients-14-04401-f002:**
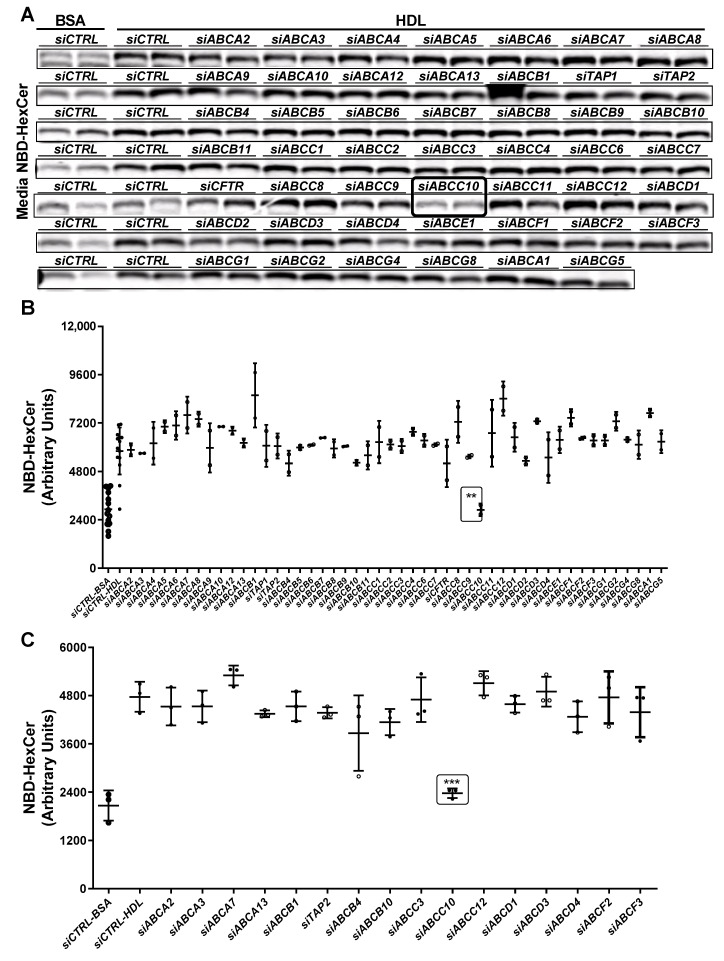
Identification of transporter(s) involved in HexCer egress to HDL. (**A**,**B**) Primary screening. Huh-7 cells were transfected in duplicate with of *siCTRL* or siRNA against various indicated membrane transporters (50 nM). After 72 h, cells were washed and incubated with 2 µM C6-NBD ceramides in DMEM containing 10% FBS for 3 h. Cells were washed again thrice with DMEM containing 0.1% BSA. Subsequently, cells were incubated with either BSA or HDL (40 µg/mL) in DMEM for 6 h. Media lipids were extracted for separation by TLC. Fluorescence in HexCer bands were photographed with the PhosphorImager (**A**) and quantified using Image J and plotted (**B**). (**C**) A secondary screening for selected transporters was performed in triplicate. Values are replicates (Mean ± SD), ** *p* < 0.01 and *** *p* < 0.001 compared with *siCTRL-HDL*.

**Figure 3 nutrients-14-04401-f003:**
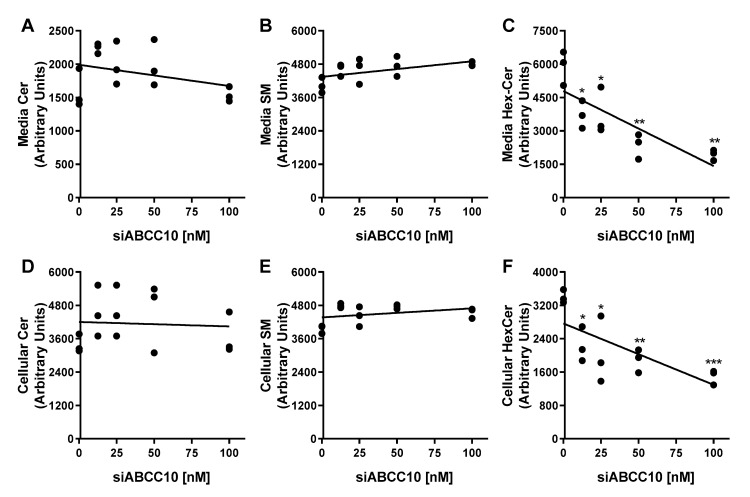
Dose response effect of *siABCC10* on HexCer efflux and synthesis. Huh-7 cells were transfected with 12.5, 25, 50, and 100 nM of either *siCTRL* or *siABCC10* in triplicate for 72 h. Cells were used to study HexCer efflux and synthesis as described in Figure 2. Lipids in the media (**A**–**C**) and cells (**D**–**F**) were extracted, separated on TLC and fluorescence in Cer, HexCer and SM bands were quantified using Image J. Values are replicates. * *p* < 0.05, ** *p* < 0.01 and *** *p* < 0.001 compared with *siCTRL-HDL* treated cells.

**Figure 4 nutrients-14-04401-f004:**
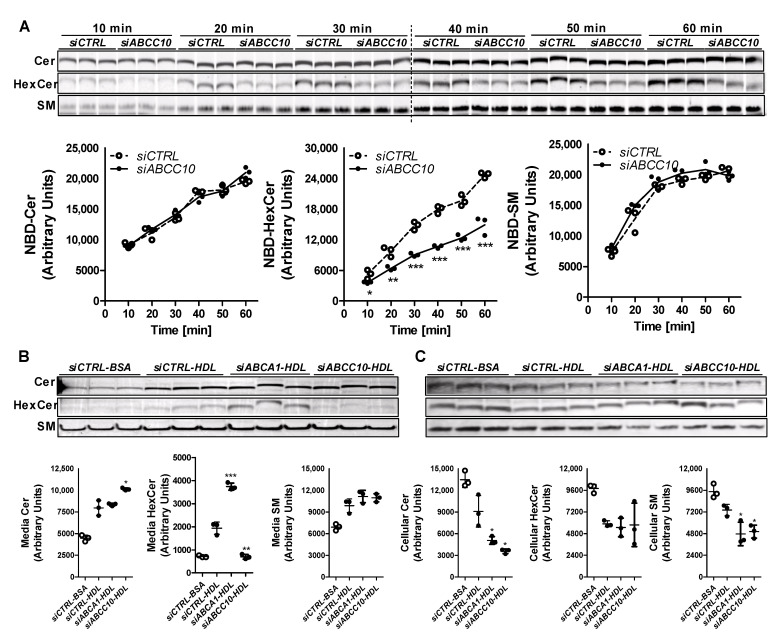
Role of ABCC10 in HexCer synthesis. (**A**) Synthesis of HexCer is decreased after treatment with *siABCC10*. Huh-7 cells were transfected (*n* = 3) with 50 nM *siCTRL* or *siABCC10.* After 72 h, cells were labeled with 2 µM C6-NBD Cer for indicated time points and were washed thrice with 0.1% BSA containing DMEM. Cellular lipids were extracted, separated on TLC plates, fluorescence in Cer, SM, and HexCer bands were scanned using the PhosphorImager (**top**), and quantified by Image J (**bottom**). Values are plotted as replicates (Mean ± SD), * *p* < 0.05, ** *p* < 0.01 and *** *p* < 0.001 compared to *siCTRL* treated cells. (**B**,**C**) ABCC10 knockdown diminishes synthesis of [^14^C]-HexCer and its transport to HDL. Huh-7 cells were transfected (*n* = 3) with 50 nM *siCTRL*, *siABCA1* or *siABCC10.* After 72 h, cells were labeled with 0.2 µCi of [^14^C]-Cer in DMEM and used for efflux studies. Media (**B**) and cell (**C**) lipids were separated using TLC and radioactive Cer, SM, and HexCer bands were visualized with the PhosphorImager (**top**) and quantified using Image J (**bottom**). Values are replicates (Mean ± SD). * *p* < 0.05, ** *p* < 0.01 and *** *p* < 0.001 compared to *siCTRL-HDL*.

**Figure 5 nutrients-14-04401-f005:**
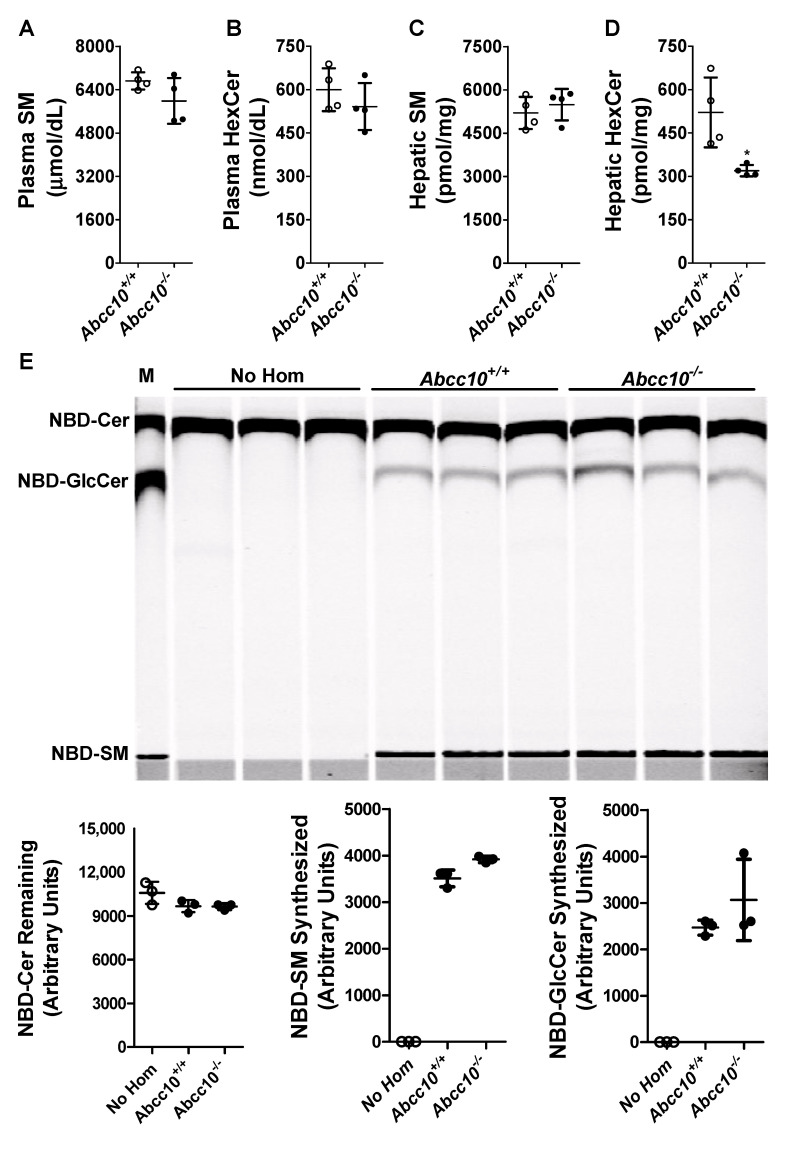
ABCC10 in hepatic HexCer metabolism. (**A**–**D**) Deletion of *Abcc10* gene in mice reduces liver HexCer levels. Overnight fasted *Abcc10^+/+^* and *Abcc10^−/−^* mice (*n* = 4, 12-week-old, chow fed) were used to measure plasma (**A**,**B**) and liver (**C**,**D**) SM and HexCer levels. Values are replicates (Mean ± SD). * *p* < 0.05 compared with *Abcc10^+/+^* mice. (**E**) Ablation of ABCC10 does not affect glucosylceramide synthase activity in liver homogenates. *Abcc10^+/+^* and *Abcc10^−/−^* liver homogenates (500 µg protein) were incubated (*n* = 3) with phosphatidylcholine (100 µg/mL), C6-NBD ceramides (3.3 µg/mL) and UDP-glucose (500 µM) at 37 °C for 1 h. As a control, reaction was also carried out in triplicate without the addition of any protein homogenates (No Hom). Lipids were extracted from the reaction mixtures, separated on TLC, fluorescence in Cer, SM, and GlcCer bands were scanned using PhosphorImager (**top**), and quantified by Image J (**bottom**). Values are plotted as replicates (Mean ± SD). In lane “M”, NBD-labeled lipids were used to identify different sphingolipids.

**Table 1 nutrients-14-04401-t001:** Levels of sphingomyelin and hexosylceramide species in the plasma and liver of wild type and ABCC10 knockout mice.

Species	Plasma (nmol/dL)			Liver (pmol/mg Protein)		
	*Abcc10^+/+^*	*Abcc10^−/−^*	*p* Value	*Abcc10^+/+^*	*Abcc10^−/−^*	*p* Value
Sphingomyelin						
C14-SM	15.4 ± 1.7	16.4 ± 1.3	0.386	0.87 ± 0.43	0.63 ± 0.36	0.425
C16-SM	2799 ± 253	2579 ± 362	0.358	1120 ± 143	1015 ± 156	0.359
C18_1-SM	76.9 ± 9.5	61.4 ± 12.3	0.093	6.42 ± 0.77	6.12 ± 0.55	0.549
C18-SM	174 ± 23	148 ± 13	0.097	157 ± 21	130 ± 17	0.093
C20_1-SM	22.4 ± 0.7	23.3 ± 6.0	0.776	4.97 ± 1.06	7.13 ± 0.85	0.019 *
C20-SM	76.6 ± 9.4	83.9 ± 13.0	0.398	153 ± 11	168 ± 18	0.205
C22_1-SM	238 ± 24	266 ± 59	0.413	109 ± 22	163 ± 20	0.011 *
C22-SM	769 ± 148	842 ± 156	0.523	1406 ± 178	1620 ± 208	0.169
C24_1-SM	1881 ± 281	1388 ± 189	0.027 *	1112 ± 189	1284 ± 152	0.206
C24-SM	664 ± 78	575 ± 88	0.181	1124 ± 156	1088 ± 123	0.730
C26_1-SM	2.30 ± 0.12	1.93 ± 0.45	0.163	2.72 ± 0.46	3.21 ± 0.25	0.110
C26-SM	2.72 ± 0.33	2.50 ± 0.15	0.270	3.43 ± 0.40	3.67 ± 0.38	0.418
Hexosylceramides						
C14-HexCer	0.36 ± 0.11	0.28 ± 0.04	0.221	0.17 ± 0.03	0.13 ± 0.01	0.045 *
C16-HexCer	127 ± 29	105 ± 17	0.239	90.1 ± 5.9	57.9 ± 7.8	0.001 **
C18_1-HexCer	0.37 ± 0.09	0.24 ± 0.01	0.028 *	BQL	0.10 ± 0.07	BQL
C18-HexCer	1.59 ± 0.22	1.30 ± 0.06	0.044 *	0.85 ± 0.09	0.60 ± 0.04	0.002 **
C20_1-HexCer	BQL	BQL	BQL	BQL	BQL	BQL
C20-HexCer	4.26 ± 0.21	4.77 ± 0.34	0.043 *	3.33 ± 0.27	2.64 ± 0.12	0.003 **
C22_1-HexCer	3.40 ± 0.58	3.32 ± 0.21	0.804	1.88 ± 0.32	1.51 ± 0.12	0.074
C22-HexCer	134 ± 15	141 ± 22	0.618	165 ± 45	113 ± 7	0.063
C24_1-HexCer	192 ± 29	170 ± 26	0.302	130 ± 35	67 ± 3	0.012 *
C24-HexCer	133 ± 62	110 ± 21	0.509	128 ± 36	74 ± 5	0.025 *
C26_1-HexCer	2.00 ± 1.09	1.95 ± 0.34	0.933	1.33 ± 0.32	0.97 ± 0.01	0.066
C26-HexCer	2.11 ± 0.35	2.42 ± 0.51	0.355	1.08 ± 0.15	0.83 ± 0.09	0.029 *

Mean + SD. A *p* value of < 0.05 was considered statistically significant. * *p* < 0.05 and ** *p* < 0.01. BQL, below quantitation limit.

## Data Availability

The authors confirm that the data supporting the findings of this study are available within the article.

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
