# Peer review of "ATP-Binding Cassette Transporter Family C Protein 10 Participates in the Synthesis and Efflux of Hexosylceramides in Liver Cells"

_nutrients, 2022, doi:10.3390/nu14204401_

Round 1

Reviewer 1 Report

This manuscript is a follow-up study of the author's previous works on ATP binding cassette family proteins. Herein they demonstrated that ATP-binding cassette transporter protein C10 (ABC-C10) has an effect on the synthesis of HexCer and its efflux to HDL. I hope the authors could address two minor issues.

In the Introduction section, can the authors include a figure showing the structural differences of Cer, SM, and HexCer? This would help readers not familiar with lipids to better understand this study. Table 1 is also misaligned and hard to read.

In the Discussion section, can the authors further speculate the possible ways for the ABC-C10 protein to influence HexCer synthesis. Could the protein have enzymatic activity? Or is it just transporting the necessary substrates for HexCer synthesis?

Author Response

Reviewer 1

Comments and Suggestions for Authors

This manuscript is a follow-up study of the author's previous works on ATP binding cassette family proteins. Herein they demonstrated that ATP-binding cassette transporter protein C10 (ABC-C10) has an effect on the synthesis of HexCer and its efflux to HDL. I hope the authors could address two minor issues.

In the Introduction section, can the authors include a figure showing the structural differences of Cer, SM, and HexCer? This would help readers not familiar with lipids to better understand this study. Table 1 is also misaligned and hard to read.

Response: We have now added chemical structures and illustrated conversion of ceramide to sphingomyelin and hexosylceramide. We have also aligned the Table 1. Thanks for pointing this out.

In the Discussion section, can the authors further speculate the possible ways for the ABC-C10 protein to influence HexCer synthesis. Could the protein have enzymatic activity? Or is it just transporting the necessary substrates for HexCer synthesis?

Response: We do not think ABCC10 has enzyme activity. If it had HexCer synthesis activity then we would have seen a difference in HexCer synthesis in Figure 5E (previous Figure 4E).

Reviewer 2 Report

In addition to SMs and ceramides, sugar derivatives of Cers, hexosylceramides (HexCer) are the major circulating sphingolipids. It is known that silencing of ABCA1 transmembrane protein function for instance in cases of LOF variants of ABCA1 gene results in low levels of HDL as well as a concomitant reduction in plasma HexCer levels. However, proteins involved in hepatic synthesis and egress of HexCer from cells is not well known although ABCA1 seems to be indirectly controlling the HexCer plasma levels. In this paper by Iqbal et al. the authors have verified that ABCC10 is involved both in synthesis and efflux of HexCer using Huh-7 hepatocyte cell culture conditions. The research protocol is very relevantly orchestrated and the methods are valid to test the aims of the study. In addition to in vitro cell conditions the authors have used appropriate mouse models to show the ABCC10 role in liver HexCer metabolism. There are certain issues, however, that need further discussions.

MAJOR COMMENTS

1. In dose-dependent experiments regarding the use of siABCC10,  HexCer was significantly reduced in hepatocytes and in cell culture media (Fig.2). What was the effect of this inhibition on cholesterol efflux from the cells to HDL acceptor?

2. Location of ABCC10 is in Golgi apparatus and once it is involved in enhancing synthesis of HexCer as well as HexCer efflux the question is whether ABCC10 will shuttle between Golgi region and plasma membrane of hepatocytes? What might be its membrane location in relation to site of ABCA1? Is the HexCer entry to HDL as a concerted action by ABCC10 and ABCA1? Via what mechanistic way do they possibly interact?

3. In siABCC10-HDL efflux assay (Fig.3B) HexCer efflux to HDL was reduced but not SM efflux. What was the outcome in terms of PCs?

MINOR POINT

1. Table 1; revise the numerical data that it is aligned properly.

2. Macrophages have ABCA1 expression but do they express also ABCC10 and what might be its physiological relevance if it is expressed?

3. Are there any limitations when using Huh-7 hepatocytes and what was the basis to use just this cell model?

Author Response

Reviewer 2

Comments and Suggestions for Authors

In addition to SMs and ceramides, sugar derivatives of Cers, hexosylceramides (HexCer) are the major circulating sphingolipids. It is known that silencing of ABCA1 transmembrane protein function for instance in cases of LOF variants of ABCA1 gene results in low levels of HDL as well as a concomitant reduction in plasma HexCer levels. However, proteins involved in hepatic synthesis and egress of HexCer from cells is not well known although ABCA1 seems to be indirectly controlling the HexCer plasma levels. In this paper by Iqbal et al. the authors have verified that ABCC10 is involved both in synthesis and efflux of HexCer using Huh-7 hepatocyte cell culture conditions. The research protocol is very relevantly orchestrated and the methods are valid to test the aims of the study. In addition to in vitro cell conditions the authors have used appropriate mouse models to show the ABCC10 role in liver HexCer metabolism. There are certain issues, however, that need further discussions.

MAJOR COMMENTS

  1. In dose-dependent experiments regarding the use of siABCC10, HexCer was significantly reduced in hepatocytes and in cell culture media (Fig. 2). What was the effect of this inhibition on cholesterol efflux from the cells to HDL acceptor?

Response: We did not study cholesterol efflux, therefore, we do not know whether ABCC10 could affect cholesterol efflux. Further, we are unaware of any studies linking ABCC10 to cholesterol efflux.

  1. Location of ABCC10 is in Golgi apparatus and once it is involved in enhancing synthesis of HexCer as well as HexCer efflux the question is whether ABCC10 will shuttle between Golgi region and plasma membrane of hepatocytes? What might be its membrane location in relation to site of ABCA1? Is the HexCer entry to HDL as a concerted action by ABCC10 and ABCA1? Via what mechanistic way do they possibly interact?

Response: We found two papers that have studied intracellular localization of ABCC10. The protein appears to be in plasma membrane as well as in intracellular compartments. Authors have concluded that intracellular localization of ABCC10 was unperturbed by various compounds used in the study [1,2]. We have now cited these papers in discussion. Detail immunohistochemical and subcellular fractionation studies are needed to address subcellular localization of ABCC10.

  1. In siABCC10-HDL efflux assay (Fig. 3B) HexCer efflux to HDL was reduced but not SM efflux. What was the outcome in terms of PCs?

Response: Again, we did not study the efflux of PC. Therefore, we cannot comment on the role of ABCC10 on PC efflux.

MINOR POINT

  1. Table 1; revise the numerical data that it is aligned properly.

Response: Done.

  1. Macrophages have ABCA1 expression but do they express also ABCC10 and what might be its physiological relevance if it is expressed?

Response: Yes, ABCC10 is in macrophages. We have now cited these studies in discussion. They report that ABCC10 expression is high in normal tissue compared to atherosclerotic tissue.

  1. Are there any limitations when using Huh-7 hepatocytes and what was the basis to use just this cell model?

Response: We have been using Huh-7 cells in our studies as these cells are easy to transfect. More importantly, they synthesize and secrete lipoproteins similar to HepG2 cells [3].

  1. Wang, J.Q.; Wang, B.; Teng, Q.X.; Lei, Z.N.; Li, Y.D.; Shi, Z.; Ma, L.Y.; Liu, H.M.; Liu, Z.; Chen, Z.S. CMP25, a synthetic new agent, targets multidrug resistance-associated protein 7 (MRP7/ABCC10). Biochem Pharmacol 2021, 190, 114652, doi:10.1016/j.bcp.2021.114652.
  2. Zhang, H.; Kathawala, R.J.; Wang, Y.J.; Zhang, Y.K.; Patel, A.; Shukla, S.; Robey, R.W.; Talele, T.T.; Ashby, C.R., Jr.; Ambudkar, S.V., et al. Linsitinib (OSI-906) antagonizes ATP-binding cassette subfamily G member 2 and subfamily C member 10-mediated drug resistance. Int J Biochem Cell Biol 2014, 51, 111-119, doi:10.1016/j.biocel.2014.03.026.
  3. Meex, S.J.; Andreo, U.; Sparks, J.D.; Fisher, E.A. Huh-7 or HepG2 cells: which is the better model for studying human apolipoprotein-B100 assembly and secretion? J Lipid Res 2011, 52, 152-158, doi:10.1194/jlr.D008888.